# A Novel, LAT/Lck Double Deficient T Cell Subline J.CaM1.7 for Combined Analysis of Early TCR Signaling

**DOI:** 10.3390/cells10020343

**Published:** 2021-02-06

**Authors:** Inmaculada Vico-Barranco, Mikel M. Arbulo-Echevarria, Isabel Serrano-García, Alba Pérez-Linaza, José M. Miranda-Sayago, Arkadiusz Miazek, Isaac Narbona-Sánchez, Enrique Aguado

**Affiliations:** 1Institute of Biomedical Research Cadiz (INIBICA), 11009 Cádiz, Spain; inmaculada.vicobarranco@gmail.com (I.V.-B.); mmartinezdeae@outlook.com (M.M.A.-E.); josemaria.miranda@cesurformacion.com (J.M.M.-S.); isaacnarbona@yahoo.es (I.N.-S.); 2Rheumatology Section, Unit of Orthopedic Surgery, Traumatology and Rheumatology, HUPM, 11009 Cádiz, Spain; isabelsg231@hotmail.com (I.S.-G.); alba_pl_91@hotmail.com (A.P.-L.); 3Department of Biochemistry and Molecular Biology, Wroclaw University of Environmental and Life Sciences, 50-375 Wroclaw, Poland; arkadiusz.miazek@upwr.edu.pl; 4Department of Biomedicine, Biotechnology and Public Health (Immunology), Universidad de Cádiz, 11002 Cádiz, Spain

**Keywords:** Lck, LAT, J.CaM1.6, TCR, signaling

## Abstract

Intracellular signaling through the T cell receptor (TCR) is essential for T cell development and function. Proper TCR signaling requires the sequential activities of Lck and ZAP-70 kinases, which result in the phosphorylation of tyrosine residues located in the CD3 ITAMs and the LAT adaptor, respectively. LAT, linker for the activation of T cells, is a transmembrane adaptor protein that acts as a scaffold coupling the early signals coming from the TCR with downstream signaling pathways leading to cellular responses. The leukemic T cell line Jurkat and its derivative mutants J.CaM1.6 (Lck deficient) and J.CaM2 (LAT deficient) have been widely used to study the first signaling events upon TCR triggering. In this work, we describe the loss of LAT adaptor expression found in a subline of J.CaM1.6 cells and analyze cis-elements responsible for the LAT expression defect. This new cell subline, which we have called J.CaM1.7, can re-express LAT adaptor after Protein Kinase C (PKC) activation, which suggests that activation-induced LAT expression is not affected in this new cell subline. Contrary to J.CaM1.6 cells, re-expression of Lck in J.CaM1.7 cells was not sufficient to recover TCR-associated signals, and both LAT and Lck had to be introduced to recover activatory intracellular signals triggered after CD3 crosslinking. Overall, our work shows that the new LAT negative J.CaM1.7 cell subline could represent a new model to study the functions of the tyrosine kinase Lck and the LAT adaptor in TCR signaling, and their mutual interaction, which seems to constitute an essential early signaling event associated with the TCR/CD3 complex.

## 1. Introduction

The activation of T lymphocytes constitutes a central event in adaptive immune responses against infectious pathogens or tumor cells. T lymphocytes recognize threatening foreign or tumoral antigens in the form of peptides bound to major histocompatibility complex molecules (pMHC). To do so, T cells express at their surface the T cell receptor (TCR) heterodimer, composed of TCR-α and TCR-β or TCR-γ and TCR-δ subunits, which, upon binding with their cognate antigens, trigger signaling cascades through the TCR associated CD3 subunits [1,2,3]. After activation, T lymphocytes undergo clonal expansion and differentiation into various types of effector T cells, which in turn contribute in different ways to the elimination of antigen and the establishment of immune memory. In humans, defects in the development of T cells usually result in the early occurrence of severe infections, revealing the essential role of this cell population for acquired immunity [4].

Most T lymphocytes express TCR-αβ heterodimers which associate with four transmembrane polypeptide subunits, namely, CD3-δ, -γ, -ε, and -ζ [1,2,5]. The TCR engagement by pMHC complexes expressed at the surface of antigen-presenting cells (APCs) triggers phosphorylation of immunoreceptor tyrosine-based activation motifs (ITAMs) located in the cytosolic domains of CD3 subunits by the p56^lck^ tyrosine kinase (Lck). This in turn recruits ZAP-70 tyrosine-kinase to the cell membrane. Once there, ZAP-70 is also phosphorylated and enzymatically activated by Lck, leading to the subsequent tyrosine phosphorylation of several intracellular substrates, including the LAT (Linker for the Activation of T cells) transmembrane adaptor.

LAT is an integral transmembrane adaptor protein of 36–38 kDa expressed in thymocytes, peripheral T lymphocytes, natural killer (NK) cells, mast cells, platelets, and pre-B cells [6,7]. Most of the functions of LAT are carried out by its four conserved carboxy-terminal tyrosines, which upon phosphorylation create docking sites for several cytosolic signaling effectors [6,8,9,10]. Many studies have established the crucial role of LAT in TCR signaling, showing that phosphorylation of LAT tyrosines constitutes nucleation sites for adapters and important signaling complexes that together mediate T cell activation [11,12]. One of the proteins binding to phosphorylated LAT is the adaptor Gads [13,14]. Gads is a small cytosolic adaptor composed of one SH2 and two SH3 domains and is constitutively bound via its carboxy-terminal SH3 domain to a proline-rich region in another cytosolic adaptor, SLP-76. Together they associate with phosphorylated LAT. Once this complex has been formed, PLC-γ1 interacts with SLP-76 via its SH3 domain [15]. PLC-γ1 is in turn phosphorylated and activated by Itk kinase, resulting in phospholipase activation [16,17]. Activated PLC-γ1 then hydrolyzes PIP2 to produce IP3 and DAG, with concomitant generation of Ca^2+^ flux and Erk activation [5]. Therefore, LAT coordinates the assembly of a multiprotein signaling complex referred to as ‘‘LAT signalosome’’, which links the TCR to the main intracellular signaling pathways that regulate T cell development and activation [18].

It is widely accepted that in T cells LAT is phosphorylated by activated ZAP-70 tyrosine kinase [6,10]. However, the first report about LAT function in T cells showed, through cotransfection experiments of 293T cells with a constitutively active form of Lck and LAT, that weak phosphorylation of LAT by Lck was possible [6]. More recently it has been shown that although ZAP-70 is the main kinase phosphorylating LAT tyrosine residues, Lck can weakly phosphorylate several LAT tyrosine residues [19,20]. The relationship between Lck kinase and LAT adaptor is still controversial, and some reports favor a role for both molecules in positive feedback loops [21,22]. However, other groups have found evidence to support the fact that LAT and Lck interaction has a negative regulatory effect on TCR signaling [23,24,25].

Much of the knowledge acquired about TCR signaling came from studies carried out with transformed T cell lines, and one of the best known of these model systems is the Jurkat leukemic T cell line [26]. The use of Jurkat cells has been critical for seminal contributions to the field of T cell biology, such as the finding that TCR ligation triggers intracellular Ca^2+^ mobilization [27], that cell surface expression of the TCR-αβ heterodimer requires co-expression of CD3 polypeptides [28], that TCR ligation activates PLC-γ1 [29], or the observation that TCR stimulation triggers protein tyrosine phosphorylation [30,31]. Of note, utilizing a combined selection protocol of Jurkat cells that retained TCR expression but failed to generate intracellular Ca^2+^ influx after TCR crosslinking, three independent Jurkat-derived clones designated J.CaM1, J.CaM2, and J.CaM3 were obtained [32,33,34]. Analysis of J.CaM1 (also called J.CaM1.6) cells revealed a defect in the expression of functional Lck tyrosine kinase, as a consequence of a splicing defect of Exon 7, which gives a shorter (and probably less stable) form of Lck. The introduction of the Lck cDNA in J.CaM1.6 cells restored the ability of these cells to respond to TCR stimulation [35]. Several years after the initial report, Arthur Weiss’s group found the signaling defect in J.CaM2 cells, revealing that LAT transmembrane adaptor was not expressed, and that transfection of these cells with a DNA coding for human LAT restored TCR-mediated PLC-γ1 activation, Ca^2+^ generation, and MAPK pathway activation [36]. Both cell lines have proved very useful in unraveling the molecular mechanisms that govern the activation of T cells, and the role that several domains or motifs of these proteins have in the TCR signaling cassette [8,9,35,36,37,38].

In this report, we characterize a new cell subline of J.CaM1.6 cells, originally described as being defective in Lck expression, which additionally has lost the expression of LAT. This new cell subline, that we have called J.CaM1.7, can re-express the LAT adaptor after protein kinase C (PKC) activation with PMA, in the same way as J.CaM2 cells can re-express LAT [39]. Contrary to J.CaM2, the LAT proximal promoter in J.CaM1.7 cells is similar to that of Jurkat cells. Contrary to Lck-deficient J.CaM1.6 cells, lentiviral transfection of J.CaM1.7 cells with a vector coding for human Lck is not enough to recover TCR-associated intracellular signals, and transfection of both LAT and Lck is needed for a full recovery of TCR signaling. Overall, our work shows that the new LAT negative J.CaM1.7 cell subline could represent a new model to study LAT-Lck interaction, which seems to constitute an essential early signaling event associated with the TCR/CD3 complex.

## 2. Materials and Methods

### 2.1. Antibodies and Reagents

Phorbol 12-myristate 13-acetate (PMA) and ionomycin (were purchased from Sigma-Aldrich (St. Louis, MO, USA) and were prepared in DMSO. Anti-LAT, anti-Lck, anti-PLC-γ, and phospho-Erk antibodies were from Santa Cruz Biotechnology (Heidelberg, Germany); antibodies binding ZAP-70, phospho-ZAP-70-Tyr319, β-actin, phospho-PLC-γ1-Tyr783, and phospho-LAT-Tyr171 were from Cell Signaling Technology (Danvers, MA, USA); anti-Grb2 mAb was from Abcam (Cambridge, MA, USA); and anti-Erk mAb was from BioLegend (San Diego, CA, USA). Stimulations were performed with the anti-human CD3 OKT3 monoclonal antibody (eBioscience, Thermo Fisher Scientific). Staining to assess CD3 expression on the membrane was performed with OKT3 antibody conjugated to APC (BioLegend, San Diego, CA, USA).

### 2.2. Cell Culture

The LAT-deficient J.CaM2 cell line was generously provided by Dr. Arthur Weiss, University of California, San Francisco (CA, USA). J.CaM1.6 cells were donated by Dr. Sancho, Instituto de Parasitología y Biomedicina López Neyra, CSIC, Granada, Spain. All cells were grown in complete RPMI 1640 medium supplemented with 10% FCS (both from Lonza, Basel, Switzerland), and 2 mM L-glutamine at 37 °C in a humidified atmosphere containing 10% CO_2_.

### 2.3. Cloning and Lentiviral Transduction

LAT and Lck cDNA cloning was performed as previously described [40]. Coding sequences in the plasmids were verified by sequencing and then subcloned in frame with ZsGreen in the SIN lentiviral transfer plasmid pHR’SINcPPT-Blast through site-specific recombination (Gateway LR Clonase, Invitrogen). Lentiviral supernatants were generated as previously described [40] and used to induce expression of Lck, LAT, or the bi-cistronic LAT-T2A-Lck cassette in J.CaM1.6, J.CaM1.7, J.CaM.2, or Jurkat cells. Blasticidin selection (20 μg/mL) and/or Hygromycin (500 μg/mL) was applied to transduced cells after 72 h of culture, and the expression of ZsGreen was analyzed using FACS analysis (CytoFLEX, Beckman Coulter, Brea, CA, USA).

### 2.4. Cell Viability

Seventy-two hours after transduction with lentiviral plasmids of Jurkat, J.CaM2, J.CaM1.6, and J.CaM1.7, cell viability was measured following APC Annexin V and Propidium Iodide staining protocol (BD, Pharmingen™), and analyzed by flow cytometry.

### 2.5. RNA Isolation, cDNA Synthesis and Conventional PCR

Cytoplasmic RNA was purified using TRIzol™ LS Reagent (Invitrogen, ThermoFisher Scientific, Waltham, MA, USA) according to the manufacturer’s protocol. RNA was reverse-transcribed using qScript™ cDNA Synthesis kit (QuantaBio, Beverly, MA, USA), and used in qPCR and conventional PCRs with DreamTaq DNA Polymerase (Fermentas, Thermo Fisher Scientific). Primer sequences for conventional human LAT conventional PCR were: hLATFw3, 5′ gactgccaggctcctacg 3′; hLATRv3, 5′ ctgttggcaccatcagaatcc 3′. Primer sequences for conventional human HRPT1 conventional PCR were: HRPT1Fw1, 5′ ggcgtcgtgattagtgatgatg 3′; HRPT1Rv1, 5′ gaatttatagccccccttgagc 3′.

### 2.6. Quantitative RT-PCR

The qRT-PCR assay was performed in three technical replicates per sample on a CFX Connect Real-Time PCR Detection System (Bio-Rad, Hercules, CA, USA). Amplifications were carried out with PerFeCTa^®^ SYBR^®^ Green FastMix^®^ (QuantaBio, Beverly, MA, USA). Each qRT-PCR run included 5 ng of cDNA and the reaction conditions were 3 min at 95 °C followed by 40 cycles of 15 s at 95 °C, 35 s at 60 °C, and 10 s at 72 °C. To control for the formation of primer dimers and unspecific amplification products, the melting curve analysis was employed. The relative expression of *LAT* (amplified with Fw3/Rv4 primer pair) was normalized to the expression of *HPRT1*. Each bar on the chart represents the mean ± standard error of the mean (SEM). Statistics were performed with Microsoft Excel using a two-tailed t-test. Levels of significance *p* < 0.001 are presented as *.

Primer sequences for human LAT qRT-PCR were hLATFw3, 5′ gactgccaggctcctacg 3′; hLATRv4, 5′ ccgtgtgaggccgtttgaac 3′. Primer sequences for human HRPT1 qRT-PCR were HRPT1Fw1, 5′ ggcgtcgtgattagtgatgatg 3′; HRPT1Rv2, 5′ caccctttccaaatcctcagc 3′.

### 2.7. Genomic DNA Purification and Sequencing

Genomic DNAs from Jurkat, J.CaM2, J.CaM1.6, and J.CaM1.7 cells were purified using PureLink™ Genomic DNA Mini Kit (Invitrogen, Thermo Fisher Scientific, Waltham, MA, USA). The 2208 bp LAT gene fragment ranging from −916 to +1292 was amplified with F1/R1 primer pair as previously described [39] with Phusion High-Fidelity DNA Polymerase (Invitrogen, Thermo Fisher Scientific, Waltham, MA, USA). PCR products were sequenced with the next primers: F1, 5′ gggctcggtgctatttgtaa 3′; R1, 5′ gcctgggttgtgatagtcgt 3′; F2, 5′ ccacctggtgcctacctg 3′; R2, 5′ tgaggatgtgctgtcgtagg 3′; F3, 5′ agacttcccctgccacctt 3′; R3, 5′ ttccccacacttaccaccat 3′.

### 2.8. PCR-RFLP

PCR fragments amplified with Taq polymerase (Invitrogen, Thermo Fisher Scientific, Waltham, MA, USA) using F2/R2 primer pair from Jurkat, J.CaM2, J.CaM1.6 and J.CaM1.7 genomic DNA were digested with BamHI endonuclease to confirm the presence of heterozygous g.237 C > T mutation in J.CaM2. Digestion of the wild type 443 bp PCR product generated 273 bp, 86 bp, and 84 bp fragments, whereas the mutant allele was digested into 273 bp and 170 bp fragments. F4/R4 primers were used to amplify 348 bp PCR product from J.CaM2 and Jurkat cDNA and detect heterozygous c.167 C > T change in J.CaM2. NlaIII digestion produced 362 bp, 101 bp, and 34 bp fragments from wild type DNA, whereas mutant sequence was cut into 220 bp, 141 bp, 101 bp, and 34 bp products.

### 2.9. Preparation of Cell Lysates and Western Blotting

Lentivirally transduced J.CaM2 cells were starved in RPMI 1640 without FCS for 3 h before being stimulated with anti-CD3 mAb at 37 °C. Cells were then lysed at 2.0 × 10^7^ cells/mL in 2× Laemmli buffer, followed by incubation at 99 °C for 5 min and sonication. For Western blotting, whole-cell lysates were separated by SDS-PAGE and transferred to PVDF membranes, which were incubated with the indicated primary antibodies, followed by the appropriate secondary antibody conjugated to IRDye 800 CW (Li-Cor, Lincoln, NE, USA) or horseradish peroxidase (HRP). Reactive proteins were visualized using the Odyssey CLx Infrared Imaging System (Li-Cor) or by enhanced chemiluminescence (ECL) acquired in a ChemiDoc Touch Imaging System (Bio-Rad Laboratories). For reprobing, PVDF membranes were incubated for 10 min at room temperature with WB Stripping Solution (Nacalai Tesque, Kyoto, Japan), followed by a TTBS wash.

### 2.10. Ca^2+^ Mobilization

Measurement of intracellular free Ca^2+^ was carried out using Indo-1 AM (acetoxymethyl) (2 μM; Molecular Probes, Invitrogen) as previously described [40]. Calcium measurements were performed using a Synergy MX Multi-Mode Reader (Biotek) at 37 °C. Cells were excited by light at a wavelength of 340 nm, and the fluorescence emitted at 405 and 485 nm was collected alternately per second. Calcium mobilization was evaluated by the ratio of 405/485 nm fluorescence signal.

## 3. Results

### 3.1. A New Subline of J.CaM1.6 Cells Which Do Not Express LAT

J.CaM1.6 cells grown in our laboratory under standard conditions were regularly tested for Lck expression. As previously described, J.CaM1.6 cells do not express Lck tyrosine kinase, as a consequence of a splicing defect of Exon 7, which gives a shorter (and probably less stable) form of Lck, which can be seen with polyclonal antibodies or monoclonal antibodies specific for the N-terminal part of the protein as a weaker and lower molecular weight band [35] (Figure 1A). As a control of the protein load for the analysis of Lck expression, we were using anti-LAT antibodies for the corresponding Western blots, and we found in one of those Western blots that J.CaM1.6 cells in culture had lost LAT expression (Figure 1A). Ponceau staining (not shown) and β-actin Western blot (Figure 1A) showed equal protein load in all lanes, indicating that this new cell subline was specifically negative for Lck and LAT expression. Furthermore, this new cell subline is positive for other molecules belonging to the TCR/CD3 signaling cassette, such as ZAP-70, Grb2 (Figure 1A). Quantification of three independent experiments shows no substantial differences in the expression of Grb2 or ZAP-70 among the different cell lines, while quantification of LAT and Lck shows a marked decrease of these proteins in the corresponding negative cell lines (Appendix A). We also analyzed CD3ε expression in these cells by flow cytometry, showing that the new cells express comparable levels, both in percentage and mean fluorescence intensity (MFI), to J.CaM1.6 and J.CaM2 cells, but lower than Jurkat cells (Appendix A). We called this new cell subline J.CaM1.7.

Next, to test if the failure in LAT expression of J.CaM1.7 cells was due to a post-transcriptional defect, we isolated total RNA from them, and also from the parental J.CaM1.6 cells. We used Jurkat cells as a positive control for LAT and Lck expression, and J.CaM2 cells as a negative control of LAT. After reverse transcription of total RNAs, cDNAs were obtained from J.CaM1.6, J.CaM1.7, Jurkat, and J.CaM2 cells, and amplified by PCR with specific primers for LAT and β-tubulin cDNAs. As it can be seen in Figure 1B, J.CaM1.7 cells were negative for both Lck and LAT, revealing that the failure in LAT expression was not due to a post-transcriptional defect. We also analyzed *LAT* expression by quantitative RT-PCR in the new J.CaM1.7 cell line. As it can be seen in Figure 1C, the *LAT* mRNA level of J.CaM1.6 cells was about 50% compared to Jurkat cells. It has been reported that this expression level is enough to reestablish intracellular signaling coupled to the TCR/CD3 complex [35], and fits well with the lack of phenotype of heterozygous LAT^+/−^ mice [41]. However, J.CaM1.7 cells showed 12% of LAT expression compared to Jurkat cells (Figure 1C). Although this level of expression is not as low as the one shown by J.CaM2 cells (about 2% with regard to Jurkat cells), this reduced expression is probably the reason that prevents the detection of LAT protein by Western blotting.

### 3.2. Long-Term Culture of J.CaM1.6 Cells Does Not Affect LAT Expression

Lck-deficient J.CaM1.6 cells were initially obtained by an approach in which Jurkat cells were mutagenized with ethyl methanesulfonate and were subsequently grown in the presence of PHA, which arrests cell proliferation [32]. Jurkat cells still expressing the TCR/CD3 complex and resistant to the growth-inhibitory effect of PHA might also be deficient in transmembrane signaling, and indeed authors included in their selection protocol the inability to mobilize calcium in response to anti-TCR antibody crosslinking [32]. Later on, the same group discovered that J.CaM1.6 cells were defective in the expression of functional Lck tyrosine kinase, and this defect was caused in part by a splicing defect [35]. Interestingly, transfection of a cDNA coding for Lck restored the ability of the J.CaM1.6 cells to respond to TCR stimulation [35]. Therefore, given the stochastic nature of this cell line, and the serendipity of our discovery of a new cell line double deficient in Lck and LAT expression, we wondered if J.CaM1.6 cells were prone to lose LAT expression under long periods in culture. Consequently, we kept J.CaM1.6 cells in culture for more than 6 months, and cell lysates were obtained periodically. As can be seen in Figure 2A, continuous culture of J.CaM1.6 cells at 37 °C for 28 weeks did not decrease LAT expression, and even LAT levels increased from week 20. We also analyzed the expression levels of other molecules belonging to the signaling cassette. As shown in Appendix A, no substantial changes maintained over time were observed in the expression of ZAP-70, PLC-γ, or SLP-76. From this result, we can conclude that our cells are the product of a random event which precluded LAT expression.

### 3.3. Protein Kinase C Activation Recuperates LAT Expression in J.CaM1.7 Cells

We have previously described that J.CaM2 cells are able to re-express LAT after activation of protein kinase C with phorbol 12-myristate 13-acetate (PMA) [39]. Indeed, stimulation of J.CaM2 cells with PMA for 24 h induced the expression of LAT (Figure 2B,C). In J.CaM1.6 cells (Lck deficient but LAT positive), the same PMA treatment produced an increase in the basal expression of LAT (Figure 2B,C), similar to what has been described for Jurkat cells [39]. Interestingly, similar to the LAT deficient J.CaM2 cell line, J.CaM1.7 cells recovered LAT expression after 24 h of treatment with 20 ng/mL of PMA. Calcium ionophore ionomycin abolished PMA-induced LAT expression in J.CaM1.7 cells, as well as in J.CaM2 cells, and is consistent with the previously observed negative effect of calcium signaling on the activation-induced LAT expression in Jurkat cells [42]. Thus, these results show that although LAT basal expression is strongly repressed in J.CaM1.7 cells, PKC activation stimulates LAT expression.

### 3.4. Analysis of the LAT Proximal Promoter Reveals No Changes in J.CaM1.7 Cells

As we previously did to find changes in the LAT genomic sequence of J.CaM2 cells [39], we PCR-amplified a fragment of genomic DNA obtained from J.CaM1.7 cells, comprising a region of 2208 bp surrounding the translational start site (Figure 3A, primers F1 and R1). Sequencing of this genomic region in J.CaM2 cells previously showed two heterozygous mutations, absent in Jurkat cells, and affecting different alleles. Both mutations were C to T mutations, the first one located in intron 1 at gene position 237, and the second one located in exon 4 at position 167, causing the LAT threonine 56 residue to be substituted by a methionine (Figure 3A). However, sequencing of the same region amplified from genomic DNA obtained from J.CaM1.7 cells showed no differences with the published sequence of Jurkat E6 cells (Figure 3A). The sequence of the genomic DNA obtained from J.CaM1.6 cells was also analyzed, which showed that the sequence was identical in this region to that of Jurkat cells (data not shown). The absence of those mutations was additionally confirmed by PCR–restriction fragment length polymorphism (RFLP) performed on genomic DNA from J.CaM1.7, J.CaM1.6, J.CaM2, and Jurkat cells as a control (Figure 3B).

### 3.5. Effect of LAT Expression on Cell Viability

Although kinases are typically associated with a protective function against apoptosis, it has been shown that the tyrosine kinase Lck is associated with apoptosis induced by ionizing radiation or cytotoxic drugs [43,44]. J.CaM1.6 cells, deficient in Lck expression, show decreased apoptosis in response to ionizing radiation when compared to Jurkat cells or Lck-transfected J.CaM1.6 cells, although similar cell death rates are observed after Fas engagement [43]. Similar results were obtained when the apoptosis in response to anticancer drugs was analyzed, with J.CaM1.6 cells showing strongly diminished cell death rates after treatment with fluorouracil, doxorubicin, or paclitaxel [44]. Therefore, it was of interest to analyze if LAT expression affects cell viability in J.CaM1.6 and J.CaM1.7 cells. To do so, we generated two lentiviral plasmids in order to reintroduce Lck or LAT and Lck (Figure 4A). To distinguish between the endogenous and the transfected forms, we introduced in the coding region of LAT an Avi-Tag, which would allow us to detect the transfected form by means of a specific monoclonal antibody, and also because this construct generated a form of LAT of greater size than the of endogenous one (Figure 4A). Similarly, a construct was generated including the coding region of human Lck and a 6-His tag at the C-terminal end. To express at the same time LAT and Lck, a lentiviral vector was generated with a T2A sequence between the coding regions of LAT and Lck (Figure 4A). Several viruses use 2A peptides to mediate protein cleavage, and 2A motifs can be used to generate multi-cistronic expression vectors [45]. We introduced a T2A sequence because this sequence has already been used to successfully express CD3-ε, CD3-γ, and CD3-δ. Both plasmids also contained an IRES sequence followed by the coding region of the fluorescent protein ZsGreen, which would allow us to verify the level of lentivirally transduced cells by flow cytometry.

To check whether these constructs allowed the expression of these proteins, Hek Lenti-X cells were transfected with these plasmids, and the level of transfection and expression of LAT and Lck was analyzed. As can be seen in Figure 4B, both plasmids transfected HEK cells, with the efficiency exceeding 95% of cells expressing ZsGreen. Moreover, HEK cells transfected with the plasmid containing the coding region of Lck allowed the expression of this tyrosine kinase, while cells transfected with the plasmid coding for LAT and Lck expressed both proteins (Figure 4C). Of note, the LAT immunoblot shows that Hek cells transfected with the vector coding for LAT and Lck with a T2A sequence also expressed the LAT/Lck fusion protein (Figure 4C, black arrow), although the amount of this protein present was significantly less than that of the individual proteins.

Next, we lentivirally transduced J.CaM1.6 and J.CaM1.7 cells (Lck-deficient and double LAT-Lck-deficient, respectively) with the corresponding lentiviral expression plasmids for Lck or LAT + Lck. Although the bi-cistronic plasmid did not allow expression of LAT and Lck for long periods, the transient expression of both proteins did allow us to evaluate the effect on cell viability, with similar levels of transfection in both types of cells with the lentiviral plasmid coding for Lck or the one coding for LAT and Lck (Figure 5A). Jurkat cells and LAT deficient J.CaM2 cells were also lentivirally transduced with both plasmids as controls. Seventy-two hours after lentiviral transduction cell viability was assessed by means of Annexin V and Propidium Iodide staining, and no differences could be detected between Lck-deficient J.CaM1.6 cells and the double-deficient LAT-Lck J.CaM1.7 cells (Figure 5B). Lentiviral transduction of Lck or LAT + Lck induced a modest decrease in cell viability, which was similar in J.CaM1.6 and J.CaM1.7 cells, and was also similar to what occurred in J.CaM2 cells (Figure 5B). Lentiviral transduction of Jurkat cells did not affect their viability. Therefore, the loss of LAT expression that has occurred in J.CaM1.7 cells does not seem to affect their viability, when compared to J.CaM1.6 cells, nor when LAT or LAT + Lck are reintroduced in both cell types.

### 3.6. Simultaneous Expression of LAT and Lck Is Needed to Recover TCR Signaling in J.CaM1.7 Cells

As commented above, long-term culture of cells transduced with the bi-cistronic lentiviral vector for LAT + Lck with the T2A sequence was not possible, as after one week of culture the proliferation of these cells was blocked (both J.CaM1.6 and J.CaM1.7 cells), and it was not possible to perform further experiments (data not shown). As there were no differences in this response between both cell types, the underlying reason may be the large size of the lentiviral vector used to simultaneously express LAT and Lck. Therefore, we generated a new lentiviral vector for a fusion protein containing the coding region of human LAT and a 6-His tag at its C-terminal end (Figure 6A). Similar to the Lck lentiviral plasmid, this plasmid also contained an IRES sequence followed by the coding region of the fluorescent protein mCherry which would allow us to verify the level of lentivirally transduced cells by flow cytometry. We lentivirally transduced J.CaM1.6 and J.CaM1.7 cells with the expression plasmid for Lck or with a combination of the LAT and Lck expression plasmids. We also transduced Jurkat and J.CaM2 cells as controls. As can be seen in Figure 6B, transduction of J.CaM1.6 and J.CaM1.7 cells with the Lck plasmid reconstituted the expression of this tyrosine kinase, and double transduction with LAT and Lck plasmids recovered LAT and Lck expression in J.CaM1.7 cells.

Next, we analyzed calcium responses after anti-CD3 stimulation. Indo-1AM-labeled cells were stimulated with 1 μg/mL of OKT3 mAb and Ca^2+^ influx was analyzed. As previously reported [35], Lck-deficient J.CaM1.6 cells were unable to increase intracellular Ca^2+^ levels in response to CD3 stimulation, but lentiviral transfection of Lck or the double transfection of LAT and Lck recovered calcium signaling (Figure 6C, left panel). However, the reintroduction of Lck in J.CaM1.7 cells was not sufficient to recover calcium signals after stimulation with anti-CD3 mAb, while the double transfection with LAT and Lck recovered the calcium increase in a similar way as it happened in J.CaM1.6 cells (Figure 6C, right panel).

Then, to check other intracellular signals associated with the TCR/CD3 complex, we performed a series of experiments in which the different cell types were stimulated with OKT3 during different periods, and the cell lysates were analyzed for the phosphorylation of PLC-γ, Erk and LAT-Y171 using Western blot. As it can be seen in Figure 7, both non-transduced J.CaM1.6 and J.CaM1.7 cells did not show phosphorylation of PLC-γ after CD3 stimulation. As expected, re-expression of Lck was enough to recover ZAP-70 phosphorylation in both J.CaM1.6 and J.CaM1.7 cells (Figure 7). Reintroduction of Lck in J.CaM1.6 cells was sufficient to restore PLC-γ activation, but in J.CaM1.7 cells simultaneous transfection with the plasmids coding for LAT and Lck was required for the recovery of the phosphorylation of this kinase (Figure 7 and Appendix A). We also analyzed Erk phosphorylation in Lck and LAT + Lck transduced cells. As it can be observed in Figure 7, the transfection of Lck was enough to recuperate Erk phosphorylation upon CD3 stimulation, and the reintroduction of LAT in J.CaM1.7 cells was not necessary for Erk activation. Next, to confirm that LAT was properly transducing intracellular signals coming from the TCR/CD3 complex, we analyzed LAT phosphorylation in non-transduced and lentivirally transduced J.CaM1.6 and J.CaM1.7 cells. As can be seen in Figure 7, the reintroduction of Lck expression in J.CaM1.6 cells was sufficient to recover the phosphorylation of LAT tyrosine 171, and the double transfection with Lck and LAT plasmids generated a double band of phosphorylated LAT, which corresponded to the endogenous and transfected forms. As expected, only double transfection of J.CaM1.7 cells with LAT and Lck allowed the phosphorylation of LAT tyrosine 171 (Figure 7 and Appendix A).

## 4. Discussion

T lymphocytes are central players in immune responses, and the time-regulated activation of these cells is crucial to prevent immune-based diseases. After recognition of a specific antigen by the TCR receptor, T lymphocytes trigger an intracellular signal cascade that leads to the activation, proliferation, and differentiation of T cells into different types of effector cells [5]. Many of the key components of the TCR signaling cassette have now been identified and characterized, and very often these advances have been made through the use of tumor cell lines. One of the most popular is the human leukemic Jurkat cell line (for a review see in [26]). Jurkat cells allowed the discovery of essential molecular events in T cell biology, such as the need of CD3 polypeptides for the expression of the αβ-TCR heterodimer at the cell surface [46], that TCR stimulation triggers a fast increase in Ca^2+^ concentration [29] or the relevance of the ζ-chain to couple the TCR to intracellular signal transduction mechanisms [47]. Of special relevance has been the use of genetically altered Jurkat T cell lines generated by genome-wide mutagenesis approaches, which allowed to prove the essential role of the tyrosine kinase Lck and the transmembrane adaptor LAT for the transduction of intracellular signals coming from the TCR/CD3 complex [35,36]. J.CaM1.6 and J.CaM2 cell lines have been extremely useful to establish the role of some domains and consensus motifs in Lck or LAT for T cell activation. In this work, we present a new cell line doubly deficient in Lck and LAT expression, which increases the array of T cell lines for the combined analysis of the molecular events taking place after TCR engagement. As this new cell line stems from a random event from the continuous culture of J.CaM1.6 cells, we have named it J.CaM1.7.

We have found the new cell line defective in LAT expression during regular Western blotting verifications of Lck expression by J.CaM1.6 cells in culture, and anti-LAT antibodies were used as loading controls. We have shown that J.CaM1.7 cells do not express LAT protein and that LAT mRNA is strongly decreased. Interestingly, analysis of LAT expression in J.CaM1.6 cells maintained in culture for 6 months did not reveal a new loss of LAT expression in J.CaM1.6 cells, suggesting that our discovery was not a likely event. This was of interest given the potential implications of LAT-mediated intracellular signaling for TCR/CD3 complex expression at the cell surface. Although we did not find substantial differences in CD3ε expression between J.CaM1.6 and J.CaM1.7 cells, it has been shown that mouse T cells expressing a LAT mutant in which tyrosine 136 (mouse ortholog of human tyrosine 132) has been mutated to phenylalanine and which has a decreased signaling ability show reduced levels of TCR/CD3 complex [48,49]. Moreover, primary mouse CD4^+^ T lymphocytes deprived of LAT express lower levels of TCR [50]. In this context, it has been shown that constitutive signals transduced through LAT are needed to maintain surface expression of the TCR/CD3 complex in Jurkat cells [51]. On the other hand, LAT expression is transiently upregulated upon TCR engagement, although intracellular signals leading to enhanced LAT transcription are not fully understood [39,52,53]. The intensification of LAT expression depends on PKC activity as inhibition of this kinase blocks the increase produced by TCR crosslinking or PMA treatment in Jurkat cells [52]. Regarding the regulation of basal LAT expression, it has been demonstrated that LAT has a 5′ proximal promoter that spans approximately 800 bp upstream of the translation start site and contains binding sites for Ets, Runx Sp1, and Sp3 transcription factors [54,55]. Interestingly, Sp1 and Sp3 knockdown or their pharmacological inhibition with mithramycin A downmodulate LAT promoter activity [39].

We have previously shown that J.CaM2 LAT-deficient cells have no sequence difference in the entire LAT promoter, and only two point mutations have been found in the gene body [39]. One of these mutations is located in intron 1 at gene position 237, and the second one is inside exon 4 at position 167. We have found no differences in the genomic sequence of the LAT gene in J.CaM1.7 cells, neither in the promoter nor the gene body, and other *cis*-elements and/or transcription factors may have been affected in J.CaM1.7 cells leading to deficient LAT expression. In this context, it is intriguing to speculate that the very same master regulator of basal LAT expression has been affected both in J.CaM1.7 and J.CaM2 cell lines. If so, future work using next-generation RNA sequencing and/or whole genome sequencing may clarify this issue. This knowledge would allow the manipulation of basal LAT expression for medical purposes (i.e., autoimmune or lymphoproliferative diseases). As the expression of LAT in J.CaM1.6 cells is decreased with respect to Jurkat cells, it is conceivable that TCR-dependent intracellular signals favor basal LAT expression. In this setting, in the same way, that LAT has a role in keeping the expression levels of the TCR complex, reciprocally, the TCR-mediated signals are needed to maintaining LAT expression. We have also demonstrated that PKC activation by PMA treatment induces LAT re-expression in J.CaM1.7 cells, in the same way as it happens in J.CaM2 cells, with ionomycin blocking the effect induced by PMA, supporting the finely regulated mechanisms controlling both basal and activation-mediated LAT transcription.

We have also analyzed the survival rates of J.CaM1.6 and J.CaM1.7 cells before and after lentiviral transfection with Lck or LAT and Lck DNAs. There is some debate on the role Lck kinase plays in T cell survival and apoptosis. Although the role of Lck in transducing activation signals leading to proliferation and survival is broadly accepted, a critical role of Lck in activation-induced cell death (AICD) by weak agonists without ZAP-70 and LAT phosphorylation has been postulated as well [56]. In that report, pharmacological inhibition of Lck or genetic abrogation of Lck expression prevented AICD of T cells. Moreover, it has been shown that null mutations in Lck abrogate Fas signaling in T cells [57]. Lck is also needed for the apoptosis induction of endothelial cells [58]. On the other side, it has been recently shown that Lck is aberrantly expressed in T-acute lymphoblastic leukemia (T-ALL) patients displaying resistance to glucocorticoid (GC) treatment, and specific Lck inhibition or *LCK* gene silencing strikingly increases GC-induced cell death of tumor cells [59]. However, we have found no differences in basal survival rates between J.CaM1.6 (Lck deficient) and J.CaM1.7 (LAT and Lck deficient). More work is needed to clarify whether LAT is needed to transduce proapoptotic or survival signals, and this new cell line constitutes an interesting tool to address some of these issues.

The main interest that the new J.CaM1.7 cell line may have is for the study of intracellular signaling associated with the TCR. In this context, we have demonstrated that, contrary to J.CaM1.6 cells, the reintroduction of Lck expression in J.CaM1.7 cells is not enough to recover intracellular signals, such as Ca^2+^ increase or PLC-γ phosphorylation. Although the new J.CaM1.7 cells express significantly higher LAT levels than J.CaM2, this amount of LAT is not sufficient to transduce activation intracellular signals from the TCR. Both LAT and Lck had to be transfected to recuperate activation signals triggered by the TCR/CD3 complex. Therefore, the J.CaM1.7 cell line represents an interesting model to address the relationship between Lck and LAT. Although it has been demonstrated that ZAP-70 is the tyrosine kinase phosphorylating LAT, both molecules have a functional link with important functional consequences. Weiss and coworkers have shown that a proline-rich motif of LAT binds to the SH3 domain of Lck, bridging ZAP-70 to LAT and so promoting LAT phosphorylation [22]. Our group has also shown that a negatively charged stretch of amino acids in LAT is involved in LAT-Lck interaction and that such interaction is needed to fine-tune intracellular signals triggered by the TCR [25]. Therefore, it is conceivable that Lck has a role in negative regulation of TCR signaling, and J.CaM1.7 could be of help to address the relevance of LAT consensus motifs and Lck domains in this negative feedback loop.

In our analysis of TCR signaling performed in lentivirally transduced J.CaM1.6 and J.CaM1.7 cells, we have not observed significant differences between Ca^2+^ responses in J.CaM1.6 cells transfected with Lck or LAT + Lck, which suggests that a threshold level of LAT expression is enough to activate biological responses in T cells. This is in line with the phenotype of heterozygous mice for LAT knockout or several strains of LAT knockins [41,48,49,60,61]. As previously commented, J.CaM1.7 cells need the reintroduction of both Lck and LAT to recover TCR signaling. However, we have observed that Erk phosphorylation is similar in the absence of LAT, and the lentiviral transfection of LAT had no substantial effect on Erk phosphorylation. This is controversial because it has been previously reported that LAT deficiency in J.CaM2 cells was needed for Erk phosphorylation [8,9,36]. Using a mouse genetic model in which LAT molecules were expressed in the thymus to permit normal thymic selection and proper CD4^+^ T cell development, Malissen and coworkers were able, using a Cre recombinase-based system, to generate CD4^+^ T lymphocytes deprived of LAT [50]. In this system, LAT-deficient CD4^+^ T lymphocytes cells showed no ERK 1 and 2 activation after TCR engagement. Moreover, a human immunodeficiency has been described in which a loss-of-function mutation of LAT generated strongly reduced numbers of T cells, with abolished Erk signaling [62]. However, other reports support our observation of LAT-independent Erk activation, as mice harboring a mutation of LAT that prevents PLC-γ binding and activation (LAT-Y136F) show normal Erk activation [49]. Besides, Samelson and coworkers have demonstrated a LAT-independent pathway by which Erk can be activated after CD3 stimulation [63]. Our results support such a LAT-independent Erk activation pathway, which could explain the lymphoproliferative disease observed in LAT-Y136F knock-in mice as the result of imbalanced Erk- and Ca^2+^-dependent signals. In summary, we present here the initial characterization of a new cell line, J.CaM1.7, double deficient in LAT, and Lck expression. This cell line may constitute a new tool to untangle molecular events related to the TCR signaling cassette and biology of T cells. Through the lentiviral transfection of J.CaM2 cells, we have recently shown that a mutation of LAT glycine 131 to aspartate increases IL-2 production in response to CD3/CD28 engagement [64]. The new J.CaM1.7 cells may be useful to confirm these data and verify the role of Lck domains or motifs in this behavior.

## Figures and Tables

**Figure 1 cells-10-00343-f001:**
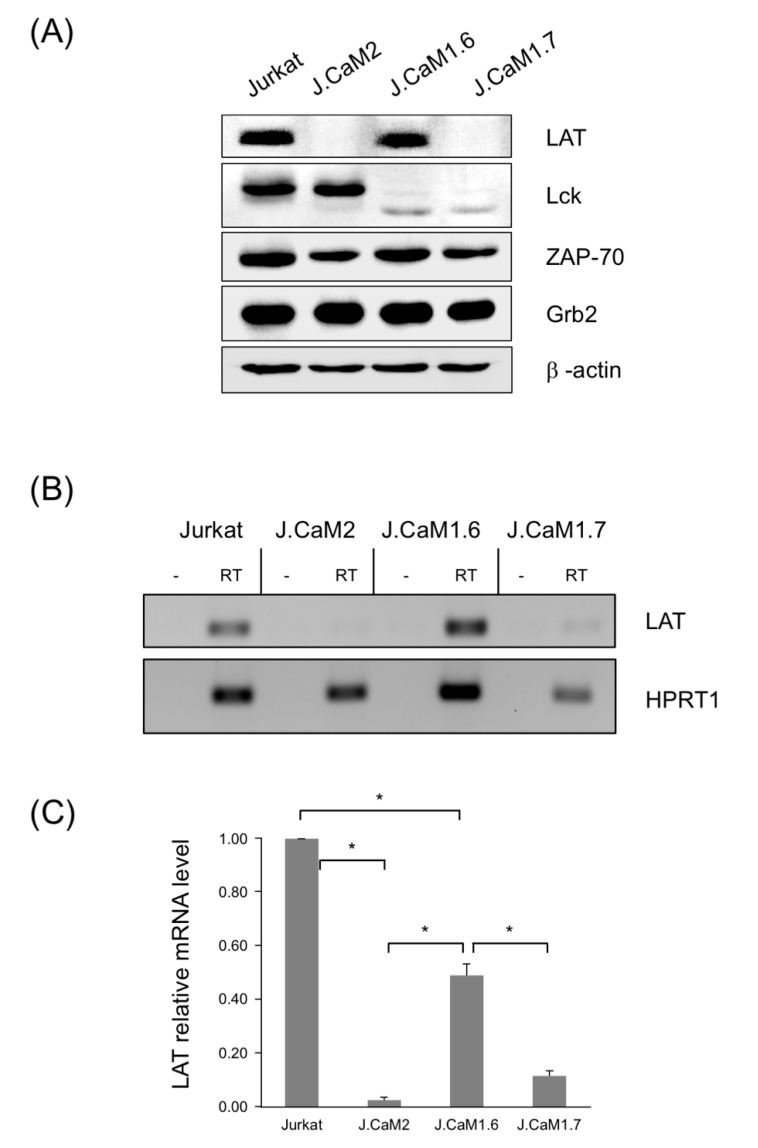
J.CaM1.7 cells are double negative for LAT and Lck expression. (**A**) J.CaM1.7 cells derived from J.CaM1.6 cells were analyzed for LAT, Lck, and β-actin expression by Western blot. Lysates obtained from 10^6^ Jurkat, J.CaM2 (LAT deficient), J.CaM1.6 (Lck deficient), and J.CaM1.7 cells were analyzed by Western blot with the indicated specific antibodies. (**B**) RT–PCR analysis of Jurkat, J.CaM2, J.CaM1.6, and J.CaM1.7 cells with specific primers for LAT and HPRT1. For each cell line, a negative control has been included in which RNA was treated without reverse transcriptase. (**C**) LAT relative mRNA levels determined by qPCR analysis in the indicated cell lines and normalized against HPRT1. Values are displayed as means ± s.d. of three independent biological replicates. Asterisks (*) represent statistical significance (*p* < 0.001).

**Figure 2 cells-10-00343-f002:**
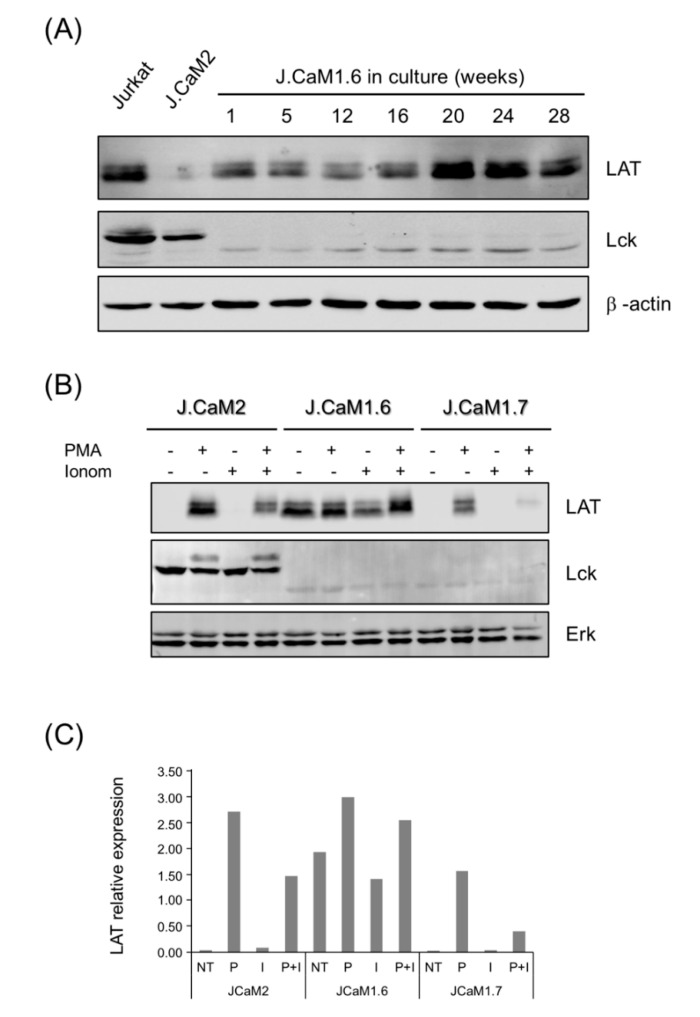
Stability and induction of LAT expression in J.CaM1.6 and J.CaM1.7 cells. (**A**) J.CaM1.6 cells were grown at 37 °C and 5% CO_2_ continuously after thawing for the indicated weeks, and then cell lysates were obtained and analyzed for LAT, Lck, and β-actin expression by Western blot. Lysates obtained from Jurkat and J.CaM2 were incorporated as positive and negative controls. (**B**) J.CaM2, J.CaM1.6, and J.CaM1.7 cells were unstimulated or PMA treated (20 ng/mL) in the absence or presence of ionomycin (Iono; 3 μM). Whole-cell lysates were prepared 24 h post-stimulation and immunoblotted with antibodies specific for LAT, Lck, and Erk. (**C**) LAT protein relative expression determined by densitometric quantification of Western blot in subfigure (**B**) normalized against Erk expression. Data in subfigures (**B**,**C**) are representative of three biological replicates. NT, not treated; P, PMA; I, ionomycin; P + I, PMA + ionomycin.

**Figure 3 cells-10-00343-f003:**
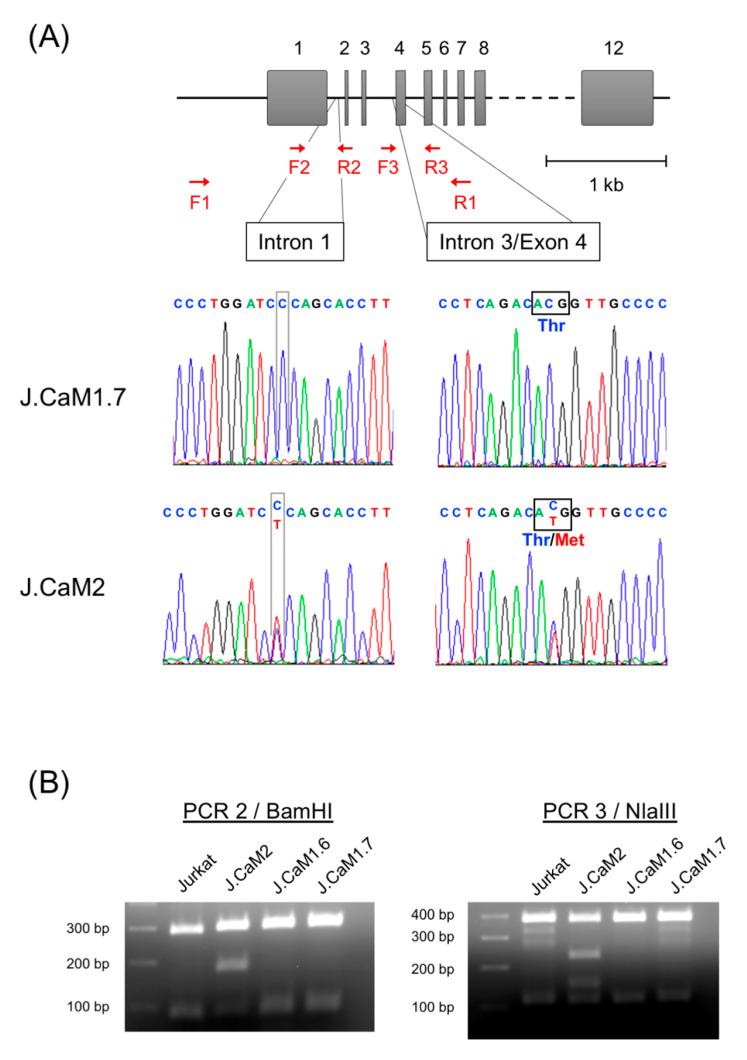
Sequence analysis in LAT gene of J.CaM1.7 cells (**A**) Schematic representation of the human genomic region of LAT gene and histograms showing homozygous wild type “C” nucleotides in intron 1 (position g.237) and exon 4 (position c.167) in J.CaM1.7 cells. As a control, the same sequence analysis was performed in J.CaM2 cells, and heterozygous variants C > T in positions g.237 and c167 are shown. F1 and R1: primers used to amplify genomic DNA to be sequenced. F2, F3, R2 and R3: primers used for sequencing. (**B**) Mutations in J.CaM2 cells were confirmed by the digestion of PCR product 2 (amplified with primers F2 and R2 from genomic DNA) with BamHI endonuclease (for the mutation in intron 1) and by the digestion of PCR product 3 (amplified with primers F3 and R3) with NlaIII. As it can be observed, only J.CaM2 cells presented both mutations, and cells J.CaM1.6 and J.CaM1.7 presented the same genotype as Jurkat cells.

**Figure 4 cells-10-00343-f004:**
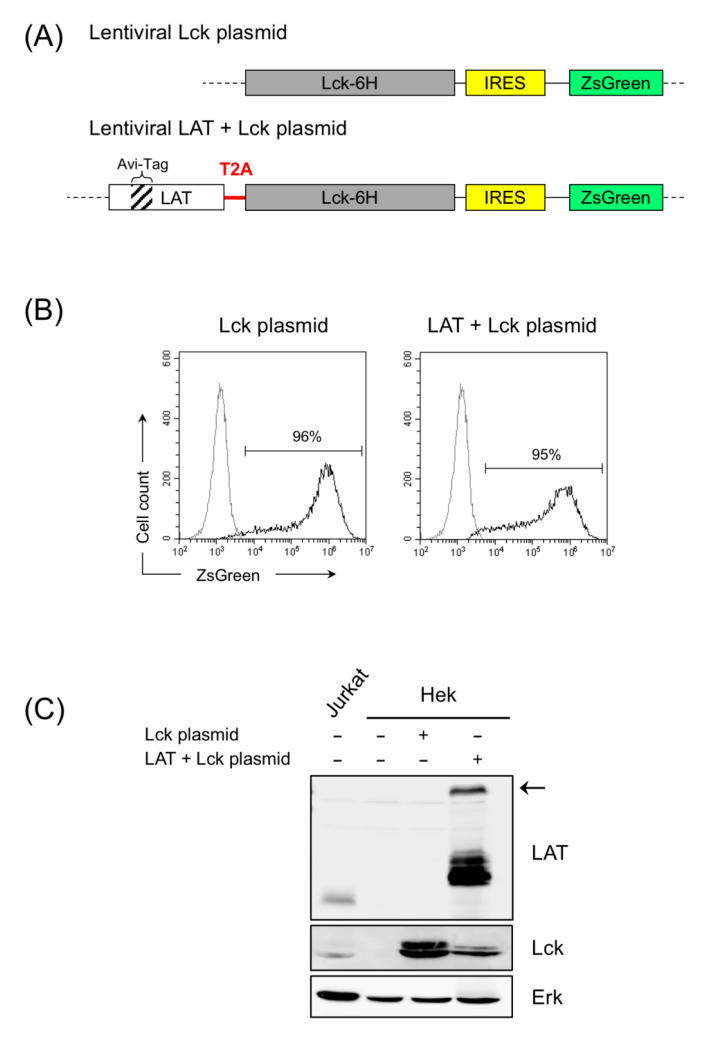
Transfection of lentiviral plasmids coding for Lck or LAT plus Lck in Hek Lenti-X cells. (**A**) Schematic representation of the coding regions included in the lentiviral plasmids generated for the expression of Lck or LAT plus Lck. The T2A sequence between LAT and Lck is indicated in red. ZsGreen reporter is located after the Lck-6His fusion gene and an IRES sequence in both plasmids. (**B**) 72 h after the transfection of the HEK Lenti-X packaging cell line, the expression of the reporter ZsGreen was determined by flow cytometry. Numbers indicate the percentage of ZsGreen positive cells (black line). Gray line corresponds to untransfected HEK Lenti-X cells. (**C**) LAT and Lck expression was analyzed in Jurkat or HEK Lenti-X cells untransfected or transfected with the indicated plasmids. Lysates obtained from 10^6^ cells were analyzed by Western blot with the indicated specific antibodies. Arrow on the right indicates the LAT-Lck fusion protein.

**Figure 5 cells-10-00343-f005:**
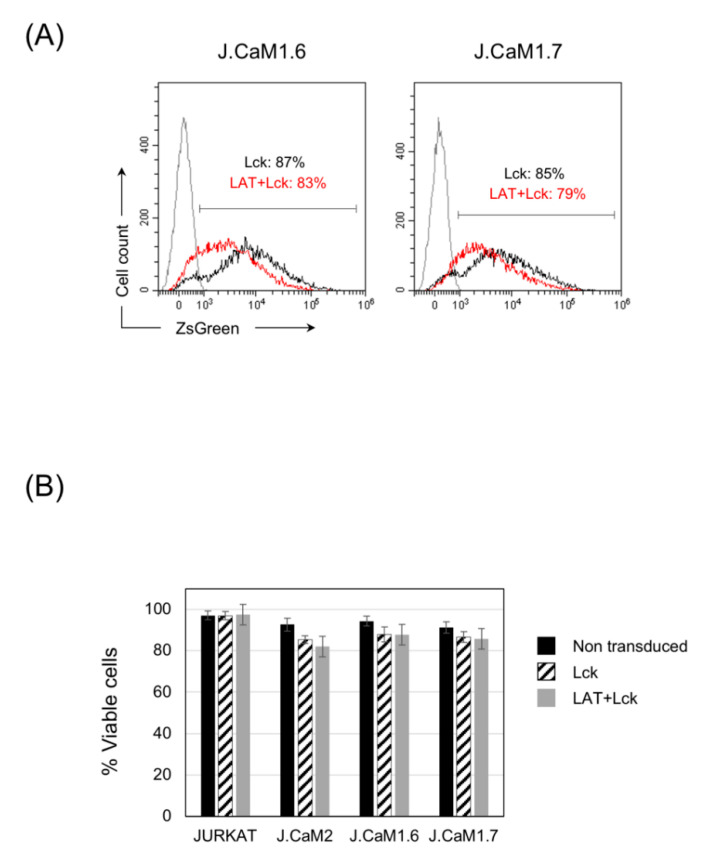
The expression of LAT or Lck does not affect the viability of J.CaM1.6 or J.CaM1.7 cells. (**A**) 72 h after lentiviral transduction of J.CaM1.6 or J.CaM1.7 cells with plasmids coding for Lck (black line) or LAT + Lck (red line) the expression of the reporter ZsGreen was determined by flow cytometry. Numbers indicate the percentage of ZsGreen positive cells. Gray line corresponds to non-transduced cells. (**B**) Viability of Jurkat, J.CaM2, J.CaM1.6, or J.CaM1.7 cells was analyzed 72 h after lentiviral transduction with the indicated plasmids using staining with APC labeled annexin V and propidium iodide (PI) and analyzed by FACS. Cell viability was estimated as the percentage of annexin V and PI negative cells. The values presented are the mean values of six experiments. Bars represent the standard error.

**Figure 6 cells-10-00343-f006:**
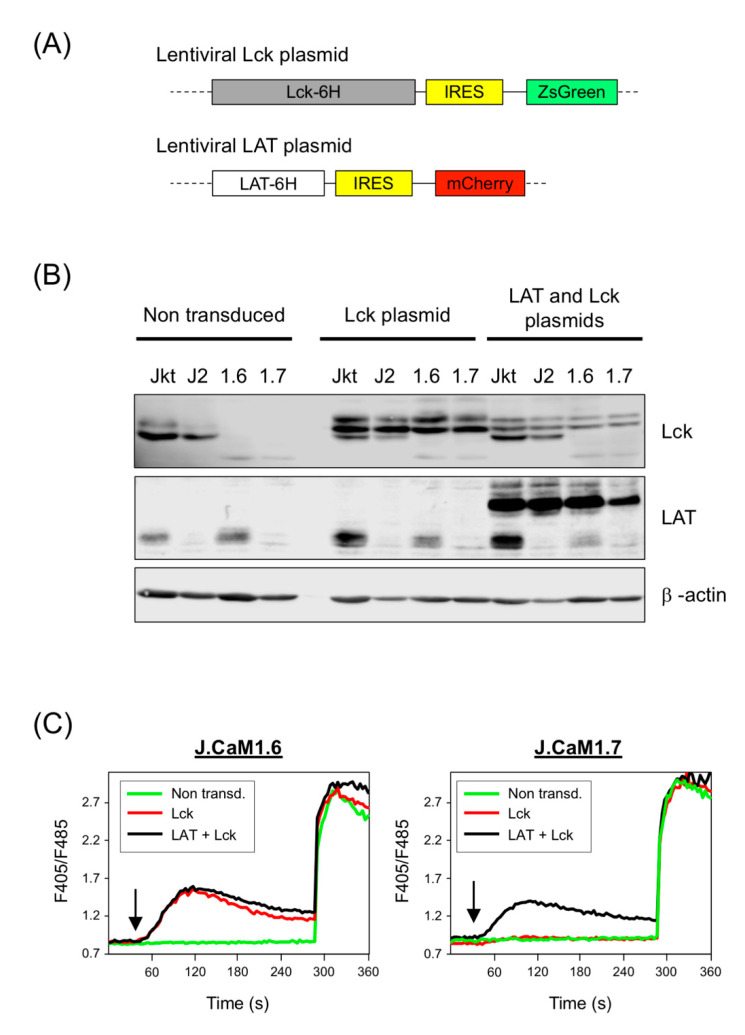
Lentiviral transduction of LAT and Lck in J.CaM1.6 and J.CaM1.7 cells and calcium analysis. (**A**) Schematic representation of the coding regions included in the lentiviral plasmids generated for the expression of Lck and LAT. ZsGreen or mCherry reporters are located after an IRES sequence, placed downstream of Lck-6His or LAT-6His, respectively. (**B**) LAT and Lck expression was analyzed after lentiviral transduction of Jurkat (Jkt), J.CaM2 (J2), J.CaM1.6 (1.6), and J.CaM1.7 (1.7) cells with plasmids coding for Lck and LAT. Immunoblots were done with the indicated antibodies. Anti-β actin immunoblot in the lower panel was performed with the same cell lysates to show equal protein loading. (**C**) J.CaM1.6 cells (left panel) and J.CaM1.7 cells non-transduced (green line) or lentivirally transduced with plasmids coding for Lck (red line) or with the two plasmids coding for LAT and Lck t were loaded with Indo-1AM and stimulated with OKT3 mAb (1 μg/mL) at the indicated times (black arrows). The intracellular Ca2^+^ concentration was determined at 37 °C through the change in Indo-1AM fluorescence. The graphics represent the average of 9 experiments.

**Figure 7 cells-10-00343-f007:**
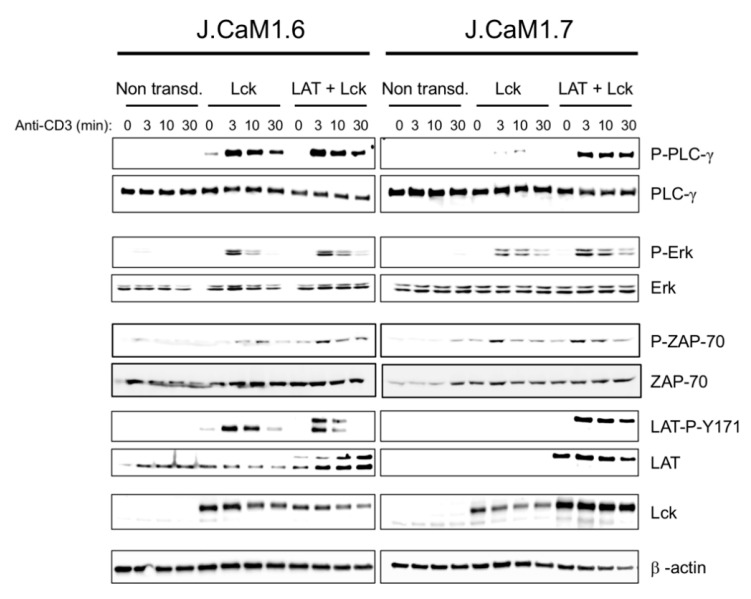
Simultaneous expression of LAT and Lck is necessary for the restoration of the intracellular signals associated with the TCR/CD3 complex in J.CaM1.7 cells. Immunoblot analysis of J.CaM1.6 (left panels) and J.CaM1.7 cells (right panels), non-transduced, or lentivirally transduced with vectors coding for Lck or LAT and Lck, as depicted. Whole-cell lysates were obtained after stimulation with an anti-CD3 antibody (OKT3, 1 µg/mL) for the indicated incubation times. β-actin immunoblots were performed to show equal protein expression. Images are representative of at least three experiments.

## Data Availability

Not applicable.

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
