# Peer review of "A Novel, LAT/Lck Double Deficient T Cell Subline J.CaM1.7 for Combined Analysis of Early TCR Signaling"

_cells, 2021, doi:10.3390/cells10020343_

Round 1

Reviewer 1 Report

Vico-Barranco et al. describe JCAM1.7, a novel subline of JCAM1.6 that has lost LAT adaptor expression. The authors identified the new subline randomly while testing for Lck and LAT expression. They demonstrate that loss of LAT expression is not caused by a post-translational defect or a mutation in the LAT proximal promoter. They show that activation-induced LAT expression is not affected in JCAM1.7 similar to JCAM2 cells. They also demonstrate that both LAT and Lck are required to recover intracellular signaling in this cell line. Finally, they propose that JCAM1.7 could be used as a new T cell model to study Lck and LAT function as well as their mutual interaction.

Even though the general conclusions of this study are correct, there are a few important points that need to be addressed:

-The authors claim that the CD3 level at the cell surface is not altered by comparing the % of CD3+ cells. But this only shows that the cells remain CD3+ and they don’t lose CD3 surface expression. In order to evaluate differences in expression levels, it would be more correct to compare the gMFI of the diverse Jurkat cell lines. In that case it seems that the CD3 level in JCAM1.7 is lower compared to all other cell lines. In addition, JCAM2 and JCAM1.6 CD3 levels are lower than Jurkat. Therefore, contrary to their conclusion, their observation fits well with what has already been published in primary cells and they should change the corresponding paragraph in the discussion. To be more precise, they should also replace CD3 by CD3ε.

-The authors studied variation of LAT expression during cell culture for the JCAM1.6 cell line and they show that this cell line not only keeps LAT expression during culture but expression levels start increasing after 20 weeks of culture. It seems to me that this increase is quite important so they should remove “slightly” from line 260. They should also add as control another protein of the early TCR signaling pathway, such as ZAP-70. More importantly, I think that they should show how LAT, Lck and ZAP-70 expression evolve during culture for all three cell lines: Jurkat, JCAM1.6 and JCAM1.7. Maybe LAT, Lck and Zap-70 levels vary a lot during cell culture for all kinds of Jurkat cell lines or not, but this should be clearly demonstrated. According to obtained results the authors should then explain potential differences or similarities between the three cell lines.

Author Response

Response to Reviewer 1

Vico-Barranco, et al. (manuscript # cells-1061409) "A novel, LAT/Lck double deficient T cell subline JCaM1.7 for combined analysis of early TCR signaling"

We appreciate the constructive suggestions of the reviewer. Below, we address, in a point by point fashion, each of his/her concerns. We believe out manuscript is substantially improved.

1 - "The authors claim that the CD3 level at the cell surface is not altered by comparing the % of CD3+ cells. But this only shows that the cells remain CD3+ and they don’t lose CD3 surface expression. In order to evaluate differences in expression levels, it would be more correct to compare the gMFI of the diverse Jurkat cell lines. In that case it seems that the CD3 level in JCAM1.7 is lower compared to all other cell lines. In addition, JCAM2 and JCAM1.6 CD3 levels are lower than Jurkat. Therefore, contrary to their conclusion, their observation fits well with what has already been published in primary cells and they should change the corresponding paragraph in the discussion. To be more precise, they should also replace CD3 by CD3ε."

We are grateful to the reviewer for his/her comments. According to his/her suggestions, we have included in Supplementary Figure 2 (Suppl. Fig 1in the first version of our manuscript) the MFI value for each cell type. Our analysis shows that J.CaM1.7 cells do not differ substantially, neither in percentage nor in MFI, from J.CaM1.6 or J.CaM2 cells. We have included a new sentence regarding this in the results section (lines 223-226). We have also changed CD3 to CD3e.

  1. - "The authors studied variation of LAT expression during cell culture for the JCAM1.6 cell line and they show that this cell line not only keeps LAT expression during culture but expression levels start increasing after 20 weeks of culture. It seems to me that this increase is quite important so they should remove “slightly” from line 260. They should also add as control another protein of the early TCR signaling pathway, such as ZAP-70. More importantly, I think that they should show how LAT, Lck and ZAP-70 expression evolve during culture for all three cell lines: Jurkat, JCAM1.6 and JCAM1.7. Maybe LAT, Lck and Zap-70 levels vary a lot during cell culture for all kinds of Jurkat cell lines or not, but this should be clearly demonstrated. According to obtained results the authors should then explain potential differences or similarities between the three cell lines."

We again thank the reviewer for his/her comments. We agree with him/her that the increase in LAT expression after 20 weeks of culture was not a small change, and concordantly, we have removed the word "slightly" from the text.

Also, following his/her suggestion, we have analyzed potential changes in the expression other proteins of the TCR signaling cassette in J.CaM1.6 cells. As it can be seen in the new Supplementary Figure 3 no substantial changes in the expression of ZAP-70, PLC-g or SLP-76 were observed over time. Consequently, we have introduced a sentence in the results section regarding this observation (lines 268-271).

Last, we also agree with the reviewer that a careful evaluation of how the expression of LAT, Lck and other proteins evolve over time in Jurkat and J.CaM2 cells would be of interest. However, this kind of analysis would take more than 6 months, making impossible to address it in the frame of this review. Moreover, although this type of study could add valuable knowledge, it is out of the scope of this work.

Reviewer 2 Report

In this paper, the authors describe a novel Jurkat mutant cell line that does not express either LCK or LAT.  They go on to characterize this line, showing that PMA treatment increases LAT expression and examining the impact of reconstitution of LCK in the presence and absence of LAT has TCR signaling. The paper is well written and the data is well presented.  However, there are several minor additions that would strengthen the conclusions of the paper.

In figure 1, the authors examine the expression of LCK, LAT, ZAP-70 and GRB2 in the various cell lines.  It would be helpful to see quantification of multiple experiments.  Also, Jurkat cells often have varying expression of signaling proteins between the mutant cell lines.  This is illustrated by the authors in figure 2A, where they interestingly show that culture of JCaM1.6 cells results in changes in LAT expression.  Have that authors looked at other critical signaling proteins, such as SLP-76, VAV1, CBL, GADS, or PLC-γ1?

In figure 4, the authors describe the use of the T2A system to express both LCK and LAT in the various Jurkat cell lines.  Could the authors show larger sections of the immunoblots to confirm that there is no expression of LAT/LCK fusion protein? 

In figure 7, the authors examine the impact of LCK or LCK and LAT expression on the TCR-induced activation of PLC-γ and ERK.  Both of these are theoretically downstream of LAT, although the authors have intriguing data to suggest ERK may be independent of LAT.  Have the authors examined pathways upstream of LAT, specifically the activation of ZAP-70?  Also, it would be helpful if the authors quantified multiple experiments for this figure.

Author Response

Response to Reviewer 2

Vico-Barranco, et al. (manuscript # cells-1061409) "A novel, LAT/Lck double deficient T cell subline JCaM1.7 for combined analysis of early TCR signaling"

We acknowledge the comments and suggestions of Reviewer 2. Below, we address, in a point by point fashion, each of his/her concerns. We believe out manuscript is substantially improved.

  1. - "In figure 1, the authors examine the expression of LCK, LAT, ZAP-70 and GRB2 in the various cell lines. It would be helpful to see quantification of multiple experiments".

We agree with Reviewer 2. Concordantly, we have performed quantifications of LAT, Lck, Grb2, ZAP-70 and b-actin immunoblots. As can be seen in the new Supplementary Figure 1, no substantial differences are observed in Grb2 or ZAP-70 expression among the different cell lines, while quantification of LAT and Lck show a marked decrease of these proteins in the corresponding negative cell lines.

  1. - "Also, Jurkat cells often have varying expression of signaling proteins between the mutant cell lines. This is illustrated by the authors in figure 2A, where they interestingly show that culture of JCaM1.6 cells results in changes in LAT expression. Have that authors looked at other critical signaling proteins, such as SLP-76, VAV1, CBL, GADS, or PLC-γ1?"

We again thank Reviewer 2 for this suggestion. We have analyzed potential changes in the expression other proteins of the TCR signaling cassette in J.CaM1.6 cells. As it can be seen in the new Supplementary Figure 3 no substantial changes in the expression of ZAP-70, PLC-g or SLP-76 were observed over time. Consequently, we have introduced a sentence in the results section regarding this observation (lines 268-271).

  1. - "In figure 4, the authors describe the use of the T2A system to express both LCK and LAT in the various Jurkat cell lines. Could the authors show larger sections of the immunoblots to confirm that there is no expression of LAT/LCK fusion protein?"

We have changed Figure 4C in the new version of our manuscript, enlarging the blot, so that the LAT/Lck fusion protein can now be seen. The expression level of the fusion protein is much lower than that of LAT, so the results of our experiments in Hek cells are still relevant. We have introduced a sentence in the results section explaining this observation (lines 351-354)

  1. - "Have the authors examined pathways upstream of LAT, specifically the activation of ZAP-70? Also, it would be helpful if the authors quantified multiple experiments for this figure"

We have performed analysis of ZAP-70 phosphorylation. As expected, re-expression of Lck was enough to recover ZAP-70 activation in both J.CaM1.6 and J.CaM1.7 cells (see the new version of Fig. 7). We agree with the reviewer that it is intriguing the LAT-independent Erk activation, and we previously discussed this point. However, it has been previously shown that CD3 stimulation can activate Erk in a LAT-independent manner (Rouquette-Jazdanian et al., Mol Cell 2012, 48: 298-312).

According to the reviewer's suggestion, we have quantified the experiments in Figure 7, and we have plotted the graphs in Supplementary Figure 4.

Round 2

Reviewer 1 Report

I think that the manuscript is significantly improved now. Some minor corrections:

-CD3 should be replaced with CD3e also in the sup.Fig.2 legend and in the sup.Fig.2 itself.

-line 453: "it" should be removed

Author Response

We appreciate the reviewer's corrections. We have already replaced CD3 with CD3ε in Supplementary Figure 2, and also in the caption to that figure. The word "it" in line 453 has also been removed.

Reviewer 2 Report

The authors have adequately addressed all my comments.

Author Response

We thank the reviewer for his/her work.